# Targeting the DNA Damage Response for Cancer Therapy

**DOI:** 10.3390/ijms242115907

**Published:** 2023-11-02

**Authors:** Ruoxi Wang, Yating Sun, Chunshuang Li, Yaoyao Xue, Xueqing Ba

**Affiliations:** 1Center for Cell Structure and Function, Key Laboratory of Animal Resistance Biology of Shandong Province, College of Life Sciences, Shandong Normal University, Jinan 250014, China; wangruoxi@sdnu.edu.cn (R.W.); 18753153871@163.com (Y.S.); 2Key Laboratory of Molecular Epigenetics of Ministry of Education, College of Life Sciences, Northeast Normal University, Changchun 130024, China; lics906@nenu.edu.cn (C.L.); xueyy459@nenu.edu.cn (Y.X.)

**Keywords:** DNA repair, DNA damage response, PARP1, double-strand DNA break repair, cancer therapy

## Abstract

Over the course of long-term evolution, cells have developed intricate defense mechanisms in response to DNA damage; these mechanisms play a pivotal role in maintaining genomic stability. Defects in the DNA damage response pathways can give rise to various diseases, including cancer. The DNA damage response (DDR) system is instrumental in safeguarding genomic stability. The accumulation of DNA damage and the weakening of DDR function both promote the initiation and progression of tumors. Simultaneously, they offer opportunities and targets for cancer therapeutics. This article primarily elucidates the DNA damage repair pathways and the progress made in targeting key proteins within these pathways for cancer treatment. Among them, poly (ADP-ribose) polymerase 1 (PARP1) plays a crucial role in DDR, and inhibitors targeting PARP1 have garnered extensive attention in anticancer research. By delving into the realms of DNA damage and repair, we aspire to explore more precise and effective strategies for cancer therapy and to seek novel avenues for intervention.

## 1. Introduction

DNA, as the most crucial genetic material in an organism, holds paramount significance for the survival and normal physiological functions of cells, which are reliant on the integrity and stability of its molecular structure. Nonetheless, DNA is not inert; it continuously experiences damage of varying degrees from both endogenous and exogenous sources, such as replication stress, telomere shortening, ultraviolet radiation, and chemical toxins, among others. Any damage incurred, if left unrepaired, can lead to mutations, genomic instability, and even cell death, contributing to numerous diseases, including cancer [1]. Consequently, cells have developed complex mechanisms, collectively referred to as the DNA damage response (DDR), to detect and repair such damage; DDR plays a pivotal role in maintaining “genomic stability”.

Extensive research on cell cultures, animal models, and human tumors has consistently demonstrated that the accumulation of DNA damage and a weakening of the DNA damage repair function both contribute to the initiation and progression of cancer, while also offering opportunities and targets for cancer therapeutics [2]. Pioneer researcher Phil Lawley, who delved into the study of DNA damage and carcinogenesis, discovered that specific alkylating agents, including dimethyl sulfate [3,4], have the ability to interact with DNA, giving rise to deleterious adducts that ultimately interfere with the regular functioning of DNA [5]. Based on this finding, a hypothesis was proposed several decades ago stating that certain cancer genes might be sensitive to these drugs; this hypothesis led to extensive research. Since then, chemotherapy drugs and radiation therapy have been found to effectively treat various cancers by inducing DNA damage.

In recent years, studies have revealed that organisms have evolved a complex network of DDR signaling pathways and repair mechanisms to safeguard genomic stability. This article aims to elucidate the key proteins involved in DDR and their roles in maintaining genomic stability and cellular homeostasis, as well as the critical mechanisms involved in cancer development. Additionally, it provides an overview of a series of important targets and small molecules identified for cancer therapy and thereby contributes to insights into the prevention and treatment of cancer diseases.

## 2. Types of DNA Damage

Research has shown that both endogenous factors (such as replication stress and oxygen radicals) and exogenous agents (such as ionizing radiation, chemotherapeutics, UV light, and polycyclic aromatic hydrocarbons) can directly or indirectly interact with DNA, leading to chemical bond breakage within DNA molecules, thereby altering the DNA’s structure and activity [6]. Diverse forms of DNA damage exist, encompassing single-strand breaks (SSBs), double-strand breaks (DSBs), base damage, DNA crosslinks, and clusters of damaged sites (Figure 1) [7,8,9]. Among these, DSBs are the most lethal. Failure to effectively and promptly repair DSBs can lead to tumorigenesis and cell death [10]. Studies have indicated that our cells undergo approximately 70,000 instances of DNA damage daily [11]. The majority of these damage instances are SSBs, which account for about 75% [12]. SSBs may arise due to oxidative damage during metabolism or base hydrolysis processes, and SSBs can also transform into DSBs [11,12]. Numerous studies have demonstrated that oxidative damage to DNA manifests as extensive damage to bases and sugars, such as modifications to guanine, including the generation of 7,8-dihydro-8-oxo-2′-deoxyguanosine (8-oxoG). Base damage is typically induced indirectly by oxidative stress due to the accumulation of reactive oxygen species (ROS) [13]. DNA crosslink is often attributed to exposure to chemical crosslinking agents, like cisplatin or ionizing radiation, which generates free radicals [14]. “Clustered damage” refers to regions where multiple types of DNA damage occur close together, presenting significant challenges for the repair processes. Clustered damage comprises base damage, apurinic/apyrimidinic (AP) site damage, and SSBs [15].

In response to the various forms of DNA damage, cells have evolved a sophisticated and intricate DDR system. DDR encompasses a range of processes, including cell cycle arrest, the regulation of DNA replication, and DNA damage repair. DDR can also influence the downstream fate of cells, such as cell death or senescence, which may depend on the immune system or occur independently of it [16,17]. The DNA damage repair pathways primarily include base excision repair (BER), which addresses single-strand breaks and base damage by excising and replacing damaged bases; nucleotide excision repair (NER), which tackles bulky DNA lesions caused by factors like UV radiation or chemical crosslinks by removing and replacing stretches of damaged DNA; mismatch repair (MMR), which is responsible for correcting mismatched base pairs, insertions, or deletions arising during DNA replication; and homologous recombination repair (HR) and non-homologous end joining (NHEJ), which collectively repair DSBs in DNA. Disruptions to DDR mechanisms have the potential to give rise to cellular transformations, mutations, or programmed cell death, thereby heightening cancer susceptibility. Consequently, comprehending the intricacies of DDR mechanisms and their functional implications is imperative in the context of cancer therapy. Recent analyses have revealed the involvement of at least 450 different proteins in the DDR process [18], providing potential targets for anticancer drug design.

### 2.1. Base Excision Repair (BER)

The BER system is a highly conserved mechanism, spanning from bacteria to humans; it is tasked with the repair of various DNA base lesions, including deamination, depurination, alkylation [19], and SSBs, all of which are crucial for maintaining genomic integrity. Any defects within the BER pathway can potentially lead to carcinogenesis; this is a well-established fact. Conversely, manipulating or altering BER mechanisms may prove to be a tool useful for survival in the face of genotoxic threats [20]. The BER pathway involves a complex interplay of numerous enzymes, as detailed in Figure 2I.

The initial enzymes in the BER pathway are DNA glycosylases, each with specificity for distinct types of DNA damage. These glycosylases primarily function by cleaving the phosphodiester bond between the deoxyribose and the incorrectly placed nitrogenous base, resulting in abasic (apurinic/apyrimidinic) sites, also known as AP sites [21]. DNA glycosylases exist in both mono-functional and bi-functional forms; mono-functional glycosylases include uracil DNA N-glycosylase (UNG), thymine DNA glycosylase (TDG), single-strand-selective mono-functional uracil-DNA glycosylase 1 (SMUG1), N-methylpurine-DNA glycosylase (MPG), and MutY homolog (MYH). Bi-functional glycosylases comprise 8-oxoguanine glycosylase 1 (OGG1), endonuclease three homolog 1 (NTH1), and Nei endonuclease VIII-like 1, 2, and 3 (NEIL1, NEIL2, and NEIL3) [22]. AP endonuclease is the next enzyme in the BER pathway. In mammalian cells, APE1 is the predominant AP endonuclease and is vital for cell survival. This enzyme functions by cleaving the phosphodiester bond in the presence of Mg^2+^ ions, generating 5′-deoxyribose phosphate (dRP) and a 3′-OH end, creating a nucleotide gap. APE1 not only possesses endonuclease activity but also functions in the oxidation-reduction activation of several transcription factors. In addition to these functions, it has proofreading capabilities and catalyzes the removal of the 3′-blocking moieties generated by bi-functional DNA glycosylases [23].

The repair process also involves DNA polymerases and DNA ligases (for nick sealing). DNA polymerases play a crucial role in the BER pathway by cleaving the blocking 5′-dRP site and further synthesizing new nucleotides at the 3′ end of the gap. As the number of nucleotides at the base site increases, two types of BER pathway exist: short-patch repair (1–2 nucleotides added) and long-patch repair (2–8 nucleotides added). For short-patch repair, Pol β conducts repair synthesis to replace the excised damaged base. For long-patch repair, after Pol β synthesizes the first nucleotide, it is released, followed by the synthesis of 2–8 nucleotides by Pol δ and ε, after which PCNA facilitates FEN1 in excising the old nucleotide strand. Following the synthesis of new nucleotides, the sealing process is carried out by the DNA ligase. DNA ligase III (Lig 3) encodes the DNA ligase responsible for forming a covalent phosphodiester bond between 3′-OH and 5′-phosphate using ATP/NAD+ as the energy source. For short-patch repair, the XRCC1–DNA ligase III complex (LIG3–XRCC1) seals the DNA. For long-patch repair, DNA ligase I catalyzes the DNA strand joining. BER is not an isolated repair pathway; rather, it is an integral part of a larger DNA damage repair mechanism. It interfaces with other repair pathways to form a network and may, in turn, be regulated by other repair pathways through feedback mechanisms [20,21,24].

In addition, research has indicated the involvement of poly (ADP-ribose) polymerase 1 (PARP1) in the BER/SSR processes. The PARP family comprises 17 members, not all of which possess polymerase activity. PARP1 rapidly recognizes and binds to AP sites and single-strand DNA breaks through its first and second zinc finger domains. Subsequently, PARP1 activation leads to PARylation modifications, during which ADP-ribose units are covalently linked to specific target proteins, with NAD+ serving as the substrate. The resulting poly(ADP-ribosyl)ation, or PARylation, carries a substantial negative charge, causing the chromatin structure to relax and facilitate the recruitment of other DNA damage repair proteins (e.g., LIG3–XRCC1) to the DNA damage site. The negatively charged, PARylated PARP1 and the DNA mutually repel each other, allowing the PARylated PARP1 to dissociate from chromatin, followed by the degradation of the PARylation, which is typically facilitated by poly(ADP-ribose) glycohydrolase (PARG) [25,26,27]. AP sites and single-strand DNA breaks serve as substrates for PARP1 activation, earning PARP1 its designation as a sensor for single-strand DNA breaks. It is worth noting that approximately 90% of PARylation is mediated by PARP1 within cells.

### 2.2. Nucleotide Excision Repair (NER)

In comparison to BER, the NER process is more intricate and is primarily employed for repairing extensive DNA adducts and DNA intra-strand crosslinks [28,29]. NER follows a stepwise mechanism involving over 30 different proteins. It operates through a “cut-and-paste” mechanism, replacing a stretch of approximately 30 nucleotides containing the lesion with the correct DNA strand. Essential contributors to NER include the seven xeroderma pigmentosum (XP) complementation groups, spanning from XPA to XPG proteins, as well as the excision repair cross-complementing group 1 protein (ERCC1), the human counterpart of yeast RAD23 (hHR23B), the replication protein A (RPA), the subunits of the transcription factor with helicase activity (TFIIH), and the Cockayne syndrome proteins A and B (CSA and CSB) [30]. Depending on the location of DNA damage within the genome, NER can be categorized into transcription-coupled repair (TCR-NER), which operates by blocking RNA polymerase elongation and repairing the damage located in actively transcribed genes, and global genome repair (GGR-NER), which addresses damage throughout the entire genome.

Apart from the initial damage recognition step, these two types are considered identical and involve five consecutive steps. The first step, as mentioned earlier, is the pivotal detection of damage, with the recognition step forming the sole distinction between TCR and GGR. In TCR-NER, the stalling of RNA polymerase by DNA damage constitutes the initial step for damage recognition. The arrested RNAPII recruits Cockayne syndrome proteins, CSB–ERCC6, which in turn recruit the CSA–ERCC8 complex. CSA and CSB recognize the damage and activate TCR-NER. In GGR-NER, the XPC–hHR23B-XPE complex can sense and identify DNA damage. It continuously surveys the genome, looking for significant DNA damage, and when it identifies the lesion, it initiates the activation of GGR-NER.

The second step involves the recruitment of the TFIIH complex to unwind the DNA helix around the damage site. TFIIH consists of two main subcomplexes: the core is composed of numerous proteins, including XPB, XPD, p62, p52, p44, p34, and p8, while the other part of TFIIH is the cdk-activating kinase subcomplex, which includes CDK7, Cyclin H, and MAT1.Notably, TFIIH possesses two ATP-dependent helicases, XPB and XPD, each with 3′-5′ and 5′-3′ helicase activities, respectively [31]. TFIIH unwinds the DNA structure, forming a bubble of approximately 30 base pairs around the damage. Two proteins, RPA and XPA, serve to stabilize the exposed DNA structure and facilitate the recruitment of the two endonucleases needed for the subsequent cleavage steps. 

Cutting the damaged strand is the rate-limiting step in the entire process. Once the pre-incision complex is prepared, it recruits the endonucleases XPG and XPF–ERCC1 to cleave the ends containing the damaged strand. These endonucleases can cut the DNA chain on the 3′ and 5′ sides of the damage site, ensuring the removal of the nucleotide-containing damaged segment. The correct positioning of XPA is of paramount importance as it is instrumental in enlisting the XPF-ERCC1 heterodimeric endonuclease. Once this is achieved, the damaged strand is released. Following this, the proteins responsible for synthesizing the missing nucleotides, namely DNA polymerases, are brought into play. DNA polymerases then fill the single-strand gap using the intact complementary strand as a template, and as a final step, DNA ligase I (LIG1) seals the 3′ nick [9,32,33] (Figure 2 II).

### 2.3. Mismatch Repair (MMR)

MMR is a highly conserved biological pathway that plays a critical role in upholding the stability of the genome. MMR’s primary function is to eliminate base pair mismatches and minor insertions/deletions that occur due to replication errors and spontaneous or induced alterations in bases (such as methylation and oxidation). This process prevents mutations from becoming permanent in dividing cells. MMR also curbshomologous recombination and has recently been found to participate in the transmission of signals related to DNA damage. MMR defects are linked to genome-wide instability, susceptibility to certain types of cancers (including HNPCC), resistance to certain chemotherapy drugs, and meiotic abnormalities and infertility in mammalian systems.

MMR bypasses the proofreading function of DNA polymerases and insertion/deletion loops (IDLs). Mismatches are recognized and initiated by the MutS-α complex, composed of the MSH2 and MSH6 proteins. MutS-α is responsible for identifying base pair mismatches and insertions/deletions ranging from single nucleotides to four-nucleotide repeat sequences; it effectively recognizes most base pair mismatches and insertions/deletions [34]. MutS-α binds to the site of damage and associates with the MutL-α complex, consisting of MLH1 and PMS2, which recruits proliferating cell nuclear antigen (PCNA) and replication factor C (RFC) to play their roles. Subsequently, DNA excision is performed by EXO1 on the DNA strand containing the erroneous base, and the gap is filled by Pol δ [35,36] (Figure 2 III).

### 2.4. Double-Strand DNA Break Repair (DSBR)

Of all the different forms of DNA damage, DSBs are the most severe. DSBs sever both strands of the DNA helix, profoundly affecting transcription, replication, and chromosome segregation. If left unrepaired, they can lead to cell death, and if repaired improperly, they can result in chromosomal translocations, which are early events in the carcinogenic process [37]. Not surprisingly, cells have evolved four distinct DSB repair pathways to address the genomic instability caused by DSBs; hundreds of different DNA repair proteins are involved in meeting this challenge. Although the mechanisms and outcomes of these pathways are highly specific, they share many common proteins and are fundamentally interdependent [38]. There are two main repair pathways for DSBs, namely, HR and the classical NHEJ [39,40].

#### 2.4.1. Homologous Recombination (HR)

HR is a relatively complex but highly accurate repair mechanism that relies on using a homologous template strand to synthesize a new one. It primarily occurs during the S and G2 phases of the cell cycle. When a DSB occurs, the MRN complex (MRE11–RAD50–NBS1 trimeric complex) is among the first to recognize and bind to the broken DNA ends [41,42]. The MRN complex is a remarkably dynamic protein assembly that interacts with various downstream HR factors. Upon binding, MRN brings in the CtBP-interacting protein (CtIP, also known as RBBP8), thereby activating the endonuclease function of the MRE11 subunit. The structure-specific endonuclease MRE11 initially binds to the DSB site and cleaves the DNA strand ending in a 5′ terminus. Subsequently, the adapter protein NBS1 assists in recruiting the PI3K kinase family member ataxia-telangiectasia mutated (ATM) to the damage site. ATM is a critical regulator in several signaling cascades. Upon DNA damage, it becomes monomeric, phosphorylating itself at Ser1981, Ser367, and Ser1893 [43,44]. Mutations in the phosphorylation sites can result in defects in the ATM signaling pathway, leading to radiation sensitivity and G2/M checkpoint abnormalities in cells. ATM activation further phosphorylates substrates like CHK2 and p53, and it phosphorylates serine residues at position 139 of the histone variant H2AX, forming γ-H2AX. MDC1 can bind to γ-H2AX and become phosphorylated by ATM, and the γ-H2AX–MDC1 complex subsequently recruits more MRN–ATM complexes to the DSB site, amplifying the damage signal throughout hundreds of kilobases of chromatin surrounding the break site [45,46,47]. Additionally, phosphorylated MDC1 can recruit E3 ubiquitin ligases RNF8 and RNF168, leading to ubiquitination modifications of histones H2A and H1, among others [48].

This chromatin ubiquitination cascade ultimately recruits the heterodimeric BRCA1-associated RING Domain 1 (BRCA1–BARD1) protein complex. BRCA1–BARD1 plays a pivotal role in the 5′ end resection, directing DSB repair towards HR. BRCA1–BARD1 interacts with CtIP and MRN in a cell-cycle-dependent fashion, stimulating MRE11 activity and initiating resection, generating 3′ single-stranded DNA (ssDNA). The ssDNA is rapidly coated by the single-strand binding protein RPA to prevent tangling. To initiate the final steps of HR, RPA must be displaced. Recombinase enzymes like RAD51, facilitated by BRCA2, replace RPA, forming a nucleoprotein filament that searches for and invades the homologous strand. DNA polymerases then synthesize a new strand using the invaded homologous strand as a template, precisely repairing the damaged DNA. After DNA repair synthesis, the extended invading strand is resolved, annealed, and ligated to the other end of the original DSB, effectively sealing the damaged region (Figure 2 IV) [9,36,49]. 

In addition to ATM, DNA damage can also activate the ATM and Rad3-related (ATR) protein. In the DNA damage response, ATR is activated by single-stranded DNA bound by the RPA protein [50,51]. ATR can phosphorylate CHK1 at the S317 and S345 sites, and phosphorylated CHK1 further activates WEE1 [39,40]. Activated WEE1 inhibits CDK1 by phosphorylating its tyrosine 15, suppressing the progression of mitosis [32,52]. Activation of the ATR–CHK1 signaling pathway activates the cell cycle checkpoint, repairs DNA damage during the replication phase, and maintains genome stability.

#### 2.4.2. Non-Homologous End Joining (NHEJ)

NHEJ is the simplest pathway for repairing DSBs and does not require a homologous template [53]. NHEJ does not demand homology at the broken ends and is an error-prone repair method that can occur at various stages of the cell cycle but is prevalent during the G1 phase. Repair proteins rapidly gather at the ends of the broken DNA molecules, and these ends are then ligated together with limited or no processing. The specific steps are as follows. 

The process begins with the Ku70–Ku80 complex, a heterodimer, binding to the ends of the broken DNA molecule. The Ku complex displays a strong affinity for DNA ends that are either blunt or have short single-strand overhangs. Once bound to the ends, the Ku heterodimer acts as a foundation for subsequent binding to the DNA-dependent protein kinase catalytic subunit (DNA-PKcs), resulting in the formation of the DNA-PK complex. DNA-PKcs undergoes autophosphorylation, and it phosphorylates neighboring chromatin as well as numerous downstream c-NHEJ factors. This phosphorylation event aids in their prompt recruitment and activation. Phosphorylated DNA-PKcs can recruit various repair proteins, including XRCC4. XRCC4 is a crucial partner of LIG4, enhancing its enzymatic activity. The DNA ligase IV–XRCC4 (LIG4–XRCC4) complex efficiently and rapidly seals blunt DNA ends or ends with short homologous overhangs. Additionally, two proteins, XRCC4-like factor (XLF, also known as Cernunnos) and the paralogue of XRCC4 and XLF (PAXX), interact with the LIG4–XRCC4 complex, playing a scaffolding role that helps to correctly position the DNA ends before ligation (Figure 2 IV). 

However, the broken DNA ends often do not have homology or contain modified nucleotides, requiring processing before ligation, typically through end trimming. NHEJ employs nucleases, including Artemis, aprataxin and PNKP-like factor (APLF), polynucleotide kinase 3′-phosphatase (PNKP), tyrosyl DNA phosphodiesterase 1 (TDP1), and aprataxin, to perform this trimming. In addition to nucleases and end-processing enzymes, c-NHEJ also employs two DNA polymerases belonging to the X family: DNA polymerase λ and DNA polymerase µ. These polymerases can add nucleotides to the 3′ end of the broken DNA, in either a templated or non-templated manner, until a ligatable end is achieved. Typically, after the action of these enzymes, the broken ends may undergo a slight reduction or an addition of a few nucleotides, resulting in microdeletions or microinsertions. Furthermore, NHEJ processes require nuclear proteins to regulate chromatin structure, allowing XRCC4 to enter the break site [36,54,55,56].

#### 2.4.3. Alternative End-Joining (alt-EJ)

In addition to HR and NHEJ, two alternative pathways for DSB repair have been identified, sharing similar mechanisms with the two main DSB repair pathways, but remaining genetically distinct. These are the alternative end-joining (alt-EJ) pathway and the single-strand annealing (SSA) pathway. Contrary to classical NHEJ, alt-EJ requires 5′ DNA end resection, which is accomplished by the MRN complex and CtIP. PARP1, a highly abundant DNA damage sensor, may promote alt-EJ by competing with the Ku heterodimer for DSB binding. Upon binding to DNA, the activation of PARP1 results in the creation of extended, negatively charged poly(ADP-ribose) (PAR) chains on PARP1 itself and on chromatin proteins in the vicinity of the break site. These chains serve as a foundation for the recruitment of subsequent DNA repair factors. [57]. PARP1 is a necessary factor recruited by DNA polymerase θ (Pol θ), which is the core mediator of alt-EJ [58]. After fill-in synthesis on both sides of the damaged region, mediated by Pol θ, the break is sealed by DNA ligase I or the DNA ligase–XRCC1 complex (LIG1/LIG3–XRCC1) (Figure 2 IV) [9,59,60,61]. 

#### 2.4.4. Single-Strand Annealing (SSA)

Compared to alt-EJ, SSA also requires 5′ end resection of the DSB ends but involves more extensive resection, because SSA can occur between longer homologous sequences located at the 3′ end of the ssDNA tails, which typically range from twenty-five to several hundred nucleotides (Figure 2 IV) [62]. These homologous sequences are typically available due to the presence of repetitive sequences at both ends of the break [38]. The extensive resection required for SSA is initiated by MRN and CtIP and extended by EXO1 and DNA2–BLM [63]. The resulting ssDNA is protected by RPA, but, unlike HR, the final steps of SSA are independent of RAD51. The annealing of complementary ssDNA regions is facilitated by the RAD52 protein, which replaces the RPA molecules bound to ssDNA. If 3′-flaps are present, they are cleaved by the XPF–ERCC1 endonuclease, ultimately completing the SSA process. This process can introduce mutations because the intervening sequences between the complementary regions are lost [38,56,64]. 

### 2.5. Fanconi Anemia Pathway

Maintaining genome stability is paramount for survival, and its failure is often associated with tumorigenesis. Consequently, there has been significant progress in understanding how cells repair various types of DNA damage to preserve genomic integrity, unveiling some unique DNA repair pathways. The Fanconi anemia pathway, also known as the FA-BRCA pathway, plays a crucial role in the repair of DNA interstrand crosslinks (ICLs) [65]. Genetic defects within this pathway result in Fanconi anemia (FA), a cancer predisposition syndrome driven by genomic instability [66]. ICLs can be induced by exogenous factors such as the chemotherapeutic agents cisplatin and mitomycin C, as well as endogenous products like aldehydes and nitrosamines. ICLs interfere with DNA replication and transcription [67]. The FA pathway is activated during the S phase, and due to functional complementation in ICL-sensitive cells, 22 FA or FA-like genes have been identified [66,68]. Among these genes, eight (FANCA, FANCB, FANCC, FANCE, FANCF, FANCG, FANCL, and FANCM) assemble into a nuclear E3 ubiquitin ligase complex known as the FA core complex, capable of monoubiquitinating the FANCD2/FANCI heterodimer (I-D heterodimer). Monoubiquitinated I-D heterodimers localize to damaged chromatin and interact with DNA repair proteins and other downstream FA proteins (FANCD1, FANCDN, FANCJ, and FANCS), facilitating repair through HR [66]. Following repair completion, the deubiquitinase ubiquitin-specific protease 1 (USP1) removes monoubiquitin from the I-D complex to shut down the network for recycling [69]. Notably, heterozygous mutations in FA genes, such as BRCA1/FANCS and BRCA2/FANCD1, increase the risk of cancer development, particularly in breast cancer [70].

## 3. Targeting DNA Damage Repair for Anticancer Therapy

Over the years, research from numerous laboratories has unveiled a highly intricate network composed of hundreds of proteins and protein complexes that identify and repair specific types of DNA damage through discrete pathways. These DNA repair pathways are meticulously coordinated with the progression of the cell cycle and operate within the immensely complex and dynamic chromatin environment. DDR signaling proteins trigger various post-translational modifications and protein complex assemblies that amplify and diversify DNA damage signals, enabling them to initiate appropriate responses. These responses can include transcriptional changes, activation of cell cycle checkpoints, selective splicing, participation in DNA repair processes, or, in the context of extensive damage, activation of cellular senescence and apoptosis pathways [71,72].

The importance of DDR is emphasized by the fact that almost all types of cancers display some degree of DNA repair deficiency [73]. Additionally, mutations in genes that encode vital DNA repair components are frequently linked to a substantially heightened vulnerability to cancer and/or premature aging [74]. Considering the pivotal role of DDR in the onset and advancement of cancer, it has become an appealing focus for the development of innovative cancer treatments, with some of them already undergoing clinical trials. The following sections will discuss key proteins in the coordination of DDR signaling events and the new drugs targeting this pathway (see Figure 3, Table 1).

### 3.1. PARP1 Inhibitors

The PARP family comprises a total of 17 members, and 90% of cellular PARylation reactions are mediated by PARP1 [75]. As previously mentioned, PARP1 plays a crucial role in DNA damage repair, and inhibitors targeting PARP1 have garnered extensive attention in anticancer research [76]. Currently, there are several PARP inhibitors available, including olaparib, talazoparib, niraparib, rucaparib, and velaparib. These inhibitors have received approval from the Food and Drug Administration (FDA) for anticancer therapy and have demonstrated significant efficacy in cancer treatment. The mechanism of action of PARP inhibitors as standalone anticancer agents continues to be actively explored.

Initially, research indicated that the mechanism through which PARP inhibitors induce cancer cell death is based on the concept of “synthetic lethality”. This means that, in the presence of *BRCA* gene deficiencies or mutations, PARP inhibitors can lead to the accumulation of DSBs, subsequently causing cell death (see Figure 3). As previously described, *BRCA* genes have long been considered integral components of the HR repair pathway. In the context of *BRCA* gene mutations, cells rely on alternative repair pathways, including those involving PARP, to repair DNA damage [75,76]. However, due to lineage mutations in *BRCA1* or *BRCA2* genes, cells become incapable of effectively repairing therapy-induced DNA double-strand breaks, leading to cell death upon PARP inhibitor treatment. Furthermore, alternative perspectives on synthetic lethality exist. When there is a substantial level of damage during the S-phase, it can trigger cell death through replication catastrophe [77]. This heightened DNA damage in S-phase cancer cells intensifies their reliance on the G2/M checkpoint. Consequently, the primary objective for the application of DDR-targeted agents in cancer therapy should, in its most basic form, focus on maximizing DNA damage during G1 and S-phase while inhibiting repair during G2, ensuring that the damage progresses into mitosis, where its effects become evident [78]. As a result, the combination of PARP1 inhibitors with drugs related to cell cycle checkpoints may hold the potential for synthetic lethality in the future.

In addition to serving as standalone anticancer agents, PARP inhibitors can also function as sensitizers in combination with other therapeutic approaches. In the 1990s and early 2000s, the development of PARP inhibitors was primarily aimed at enhancing the effectiveness of ionizing radiation and chemotherapy drugs in cancer treatment. [79,80]. Initially, PARP inhibitors were demonstrated to increase the sensitivity of cancer cells to DNA methylating agents. Subsequently, chemical sensitization effects were observed in vitro and in vivo with topoisomerase I poisons, camptothecin, temozolomide, and irinotecan. Preclinical studies demonstrated that the combination of the PARP inhibitor rucaparib with the alkylating agent temozolomide resulted in complete tumor regression in mice [81,82]. Research has also indicated that PARP inhibitors can sensitize cells to platinum-based drugs, although this effect appears to be cell-line-dependent, and chemotherapy sensitization may be due to additional toxicity against HR-deficient cells, as PARP inhibitors and platinum drugs individually induce severe cytotoxicity in HR-deficient cells [83].

The history of the clinical use of PARP inhibitors in cancer treatment is relatively short, and our understanding of the characteristics and mechanisms of resistance to PARP inhibitors in cancer is limited. At present, the majority of insights into PARP inhibitor resistance are derived from preclinical studies, particularly in vitro research. However, resistance to PARP inhibitors in clinical settings is inevitable; this is similar to the situation with other anticancer drugs. Furthermore, PARP inhibitors may be employed in the treatment of cancers with different genetic defects. Therefore, further investigation is required regarding the clinical resistance to PARP inhibitors [84].

### 3.2. ATM/ATR Inhibitor

ATM is a protein kinase that plays a pivotal role in promoting DSB repair and orchestrating the cellular response to DSBs throughout the cell cycle. ATM is primarily activated through interaction with the MRN complex, particularly its component NBS1. It serves as the principal kinase responsible for phosphorylating histone H2AX, which is a critical event that occurs rapidly following DSBs and serves as a foundational step in DNA repair mechanisms. Given its central role in DSB repair, targeting ATM for cancer therapy has garnered significant attention. The research indicates that the loss of ATM does not significantly impact sensitivity to PARP1 inhibition but robustly sensitizes cells to the inhibitors targeting the related DNA damage response kinase, ATR [85]. To date, various ATM inhibitors have been investigated for cancer treatment [86]. The first reported ATM inhibitor, KU-55933, significantly enhanced sensitivity to IR and chemotherapeutic agents that induce DSBs, such as etoposide, doxorubicin, and camptothecin [87]. Additionally, under conditions of DNA damage, KU-55933 significantly increased cell death in multiple myeloma cells [88]. Furthermore, studies have demonstrated that both the knockdown of *INPP4B* (inositol polyphosphate-4-phosphatase type II) and the use of KU-55933 can sensitize primary myeloid leukemic cells to cytarabine treatment [89]. However, due to KU-55933′s high lipophilicity, it is not suitable for in vivo use [86]. KU59403, the first ATM inhibitor developed for preclinical trials, significantly enhanced the antitumor activity of topoisomerase poisons in mice bearing human colon cancer xenografts (SW620 and HCT116), at doses that were non-toxic when administered alone and well tolerated when used in combination [90].

ATR is activated through the binding of ssDNA by RPA, which can occur due to the stalling of replication forks or during the early stages of HR following DNA end resection. ATR’s involvement in HR has led to the development of ATR inhibitors for anticancer therapy. Early ATR inhibitors lacked specificity, affecting multiple signaling pathways within cells, limiting their clinical application. Presently, several ATR inhibitors are undergoing clinical trials, including M6620 (VX-970 or berzosertib), M4344 (VX-803), AZD6738, and BAY1895344 [91,92]. The first-in-class ATR inhibitor, M6620, has shown promise both as a monotherapy and in combination with carboplatin [93]. Additionally, NU6027 has been identified as a potent inhibitor of cellular ATR activity, enhancing the cytotoxicity of hydroxyurea and cisplatin in an ATR-dependent manner. Furthermore, NU6027 has demonstrated synthetic lethality when DNA single-strand break repair is compromised, either through PARP inhibition or XRCC1 defects, in breast and ovarian cancer cell lines [94]. AZD6738 induces cell death and senescence in non-small cell lung cancer (NSCLC) cell lines. In NSCLC cell lines with intact ATM kinase signaling, AZD6738 potentiates the cytotoxicity of cisplatin and gemcitabine, while displaying potent synergy with cisplatin in ATM-deficient NSCLC cells [95].

### 3.3. DNA-PK Inhibitors

DNA-PK consists of Ku, along with a catalytic subunit of approximately 460 kDa (DNA-PKcs), and the functionality of DNA-PKcs relies on the DNA DSB binding facilitated by Ku. This complex plays a crucial role in the NHEJ pathway which is essential for effective repair by classic NHEJ. Notably, it has been observed to be upregulated in several tumor cell lines [96,97,98]. Moreover, studies have shown that thyroid cancer cells with low DNA-PKcs levels are sensitive to radiation, whereas those with high DNA-PKcs levels exhibit radiation resistance [99]. Researchers have observed that, in oral squamous cell carcinoma (OSCC) cell lines, the upregulation of DNA-PKcs following radiation treatment correlates with radiation resistance [100]. Cervical carcinoma cells surviving radiotherapy also show increased DNA-PKcs expression [101]. Targeting the phosphorylation of DNA-PKcs at T2647 with the inhibitory peptide BTW3 has been demonstrated to enhance the sensitivity of colon cancer cells to radiation therapy [102]. These findings suggest that the expression level of DNA-PKcs may serve as a promising target for cancer treatment.

Caffeine was the first compound reported to inhibit DNA-PK, and it achieves this by inhibiting DNA-PK activity through a mixed non-competitive mechanism concerning ATP [103]. Currently, several DNA-PK inhibitors are available, including NU7441, nedisertib, AZD7648, VX-984, berzosertib, ceralasertib, VX-803, BAY1895344, CC-115, NU7427, NU7026, and NU7163, among others [104,105,106]. NU7441 sensitizes breast cancer cells to ionizing radiation and doxorubicin [107]. Furthermore, the combination of NU7441 with topoisomerase inhibitors has a synergistic effect on cell proliferation in A549 cells [108]. Nedisertib, also known as M3814, enhances the effectiveness of radiation therapy in ovarian cancer animal models and non-small-cell lung cancer models [109]. AZD7648 is currently in phase I clinical trials and efficiently sensitizes cells to radiation- and doxorubicin-induced DNA damage. When combined with the PARP inhibitor olaparib, AZD7648 increases genomic instability, leading to cell growth inhibition and apoptosis [110]. VX-984 is also in phase I clinical trials and enhances the radiosensitivity of glioblastoma cells [111]. In the treatment of leukemia, NU7026 potentiates the cytotoxicity of topoisomerase II poisons [112].

### 3.4. CHK1/2 Inhibitors

The cell cycle checkpoint kinases CHK1 and CHK2 are the main downstream effectors of ATR and ATM, respectively. Studies have shown that the heightened expression of CHK1 and CHK2, leading to the activation of the DNA damage checkpoint response, can result in radioresistance. Moreover, the loss of CHK1 and CHK2 expression can reverse radioresistance both in vitro and in vivo in cells with high c-MYC expression [113]. Furthermore, enhancing CHK1 stability can promote HR-dependent DNA repair and resistance to radiation [114]. These lines of evidence suggest that CHK1/2 are promising targets for cancer therapy. There are currently several CHK1/2 inhibitors available, and many are in clinical trial stages [115].

UCN-01 (7-hydroxystaurosporine) is a first-generation CHK1 inhibitor [116]. UCN-01 has been used as a chemosensitizer [117,118], but its lack of specificity, resulting from its binding to alpha acidic glycoprotein, leads to hyperglycemia [119,120]. In addition to UCN-01, XL844 and CBP501 also serve as sensitizers, but their clinical applications have been limited due to non-specificity [121,122,123,124]. AZD7762 can simultaneously inhibit CHK1 and CHK2 [125,126], and it can be used as a standalone antitumor drug as well as a sensitizing agent [127,128,129]. Furthermore, studies have demonstrated that AZD7762 not only enhances radiation-induced apoptosis and mitotic catastrophe in p53 mutant breast cancer cells in vitro but also slows the growth of their xenografts in response to radiation in vivo [130]. The treatment of p53-deficient squamous cell carcinoma of the head and neck with AZD7762 sensitizes the cells to cisplatin through the induction of mitotic cell death [131]. Rabusertib (LY2603618) is a potent and selective small molecule inhibitor of CHK1 protein kinase activity in vitro studies, and it was the first selective CHK1 inhibitor to be introduced into clinical cancer trials [132]. LY2603618, when administered in combination with pemetrexed and cisplatin, demonstrated an acceptable safety profile [133]. Although the treatment goals were achieved, the combination of LY2603618 + pemetrexed + cisplatin will not be pursued for further development in the treatment of advanced non-squamous non-small cell lung cancer. This decision is based on concerns about the potential heightened risk of thromboembolic events associated with this combination. [134]. MK-8776 (SCH 900776) can be used both as a standalone chemotherapeutic agent and in combination with DNA antimetabolites [135], pemetrexed [136], and gemcitabine [137].

### 3.5. WEE1 Inhibitor

WEE1, functioning as a tyrosine kinase, can inhibit CDK1/2, thereby activating the G2/M cell cycle checkpoint. This leads to cell cycle arrest, preventing entry into mitosis in response to cellular DNA damage [91,138]. Studies have shown that WEE1 is overexpressed in various cancer cell types, including cervical cancers, lung cancers, breast cancers [139], squamous cell carcinoma [140], glioblastoma [141], and melanoma [142]. However, research has also indicated the loss of WEE1 expression in non-small-cell lung cancer (NSCLC), and this loss may potentially promote tumor progression [143]. These findings suggest that WEE1 could be an effective target for cancer therapy. PD0166285 is a first-generation WEE1 inhibitor, but its use is limited due to its non-selective characteristics [144,145]. Adavosertib (AZD1775) is a potent and selective small molecule inhibitor of WEE1 kinase [146]. A Phase I clinical trial was conducted to assess the safety, tolerability, pharmacokinetics, and pharmacodynamics of oral AZD1775 when administered as a monotherapy or in combination with chemotherapy, such as gemcitabine, cisplatin, or carboplatin, in patients with refractory solid tumors [147,148]. Additionally, research has found that AZD1775 can specifically enhance the sensitivity of p53-deficient tumor cells to DNA-damaging agents [146] and radiation [149]. A phase II study indicated that AZD1775 enhances carboplatin efficacy in p53-mutated tumors; however, adverse events such as fatigue, nausea, thrombocytopenia, diarrhea, and vomiting were also observed [150].

### 3.6. DNA Pol β Inhibitors

DNA Pol β plays a crucial role in both nuclear and mitochondrial BER processes [20,151,152,153]. Pol β also contributes to DSB repair through the alt-EJ pathway [154]. Furthermore, Pol β plays a vital role in the cellular life cycle, as evidenced by embryonic lethality upon knockout of the gene encoding Pol β in mice, highlighting its importance in fetal development [155]. Further research has revealed the significant role of Pol β in chemotherapy resistance, as its overexpression reduces the efficacy of anticancer drug treatments [156]. Small-scale studies across different cancer types have indicated that this enzyme is upregulated and/or mutated in many human cancers, such as colorectal cancer, where the mutation rate reaches approximately 40% [157]. This leads to decreased accuracy in DNA synthesis, leaving the genome susceptible to severe and often detrimental mutations [158]. Consequently, Pol β has been strongly regarded as a promising target for cancer therapy.

In the mid-1990s, Mizushina and colleagues began screening for small molecule inhibitors of Pol β. Their initial research focused on identifying compounds from microbial fermentation that could inhibit DNA polymerases. They discovered that linoleic acid (LA), a well-known fatty acid, could inhibit calf thymus DNA pol α and rat DNA pol β [159]. Over the next two decades, Mizushina and other researchers, along with the National Cooperative Drug Discovery Groups (NCDDGs), funded by the National Cancer Institute, identified numerous Pol β inhibitors. These included polypeptides, fatty acids, triterpenoids, sulfolipids, polar lipids, secondary bile acids, phenalenone derivatives, anacardic acid, harbinatic acid, flavonoid derivatives, and pamoic acid, among others. However, most of these inhibitors lacked sufficient effectiveness or specificity to become approved drugs [160,161]. Recently, Yuhas developed a covalent small molecule inhibitor named pro-14, which can inhibit DNA polymerase β’s binding to DNA [162]. It also synergizes with MMS and bleomycin to kill HeLa cells [162]. However, its impact on tumors requires further investigation. Therefore, future efforts should explore more extensive approaches to discover additional Pol β inhibitors with minimized side effects and enhanced potential as therapeutic agents.

### 3.7. ERCC1 Inhibitors

The NER pathway plays a crucial role in the removal of large DNA adducts induced by UV radiation, external agents, lipid peroxidation, or ROS. These adducts distort the DNA helical structure, leading to cell cycle arrest and inducing apoptosis [163,164]. Platinum-based drugs, such as cisplatin, have been developed to exploit this pathway. They crosslink DNA molecules, causing extensive DNA damage and triggering apoptosis in cancer cells. However, in many cancer cells, the NER pathway is overactive due to the overexpression of ERCC1, which can diminish the therapeutic effectiveness of cisplatin and even lead to drug resistance [165,166,167,168]. Therefore, the design of selective NER/ERCC1 inhibitors holds promise in the enhancement of the efficacy of platinum-based treatments. ERCC1 forms a heterodimer with XPF, through predominantly hydrophobic interactions within their double helix–hairpin–helix (HhH2) C-terminal regions, and this complex is at the core of NER. Additionally, this complex plays a secondary role in various SSB and DSB repair processes [169]. Thus, inhibiting the ERCC1–XPF heterodimer is an attractive target. Researchers at the University of Edinburgh employed computer-aided virtual screening (SBVS) to target pockets on the XPF binding site for ERCC1, discovering a compound capable of disrupting the XPF–ERCC1 heterodimer and impeding NER [170]. In another study, compound F06 was identified as the most potent binder; it also disrupted the XPF–ERCC1 interaction in cells [171]. Subsequent research involved optimizing F06, resulting in derivatives known as F06-4 and B5, which could inhibit the activity of the XPF–ERCC1 complex, rendering colon cancer cells sensitive to ultraviolet radiation and cyclophosphamide [172,173]. 

ERCC1 engages in specific interactions with XPA, which is a process crucial for NER [174]. Consequently, inhibiting their interaction serves to modulate the NER pathway. Researchers have employed a short peptide that mimics the binding domain of XPA with ERCC1, effectively competing with full-length XPA protein for binding to ERCC1 [175]. Additionally, based on this finding, small molecules that disrupt the ERCC1–XPA interaction have been identified, including UCN-01 [176], NERI01, and compound 10 [177]. Extensive computer screening efforts have been undertaken to identify more potent ERCC1–XPA inhibitors [178,179]. While the NER pathway has been identified as one of the most critical factors contributing to resistance to platinum-based therapy, efforts to modulate its activity have been limited. Beyond the inhibitors mentioned above, researchers have identified inhibitors of topoisomerases II and I, such as F11782 [180] and the DNA-damaging agent Ecteinascidin 743 (Et743) [181]. Et743 specifically disrupts the TCR–NER subpathway without acting as an inhibitor for any proteins involved in the NER mechanism, making it a novel class of anticancer drug with strong inhibitory activity against various tumor cells [182].

DDR is an extensive signaling network that orchestrates the recognition of DNA damage, its repair, prevention, cell cycle advancement, and cell death. This network encompasses over 450 genes encoding proteins. Therefore, in addition to the mentioned targets for designing small molecule inhibitors to treat tumors, many proteins involved in DDR can also serve as targets for inhibitor design. For example, APE1, a DNA glycosylase which recognizes and removes damaged DNA base lesions at AP sites, plays a crucial role in BER. TRC102 (methoxyamine; TRACON Pharmaceuticals) interacts with AP sites, creating AP adducts that are impervious to the action of APE1. This process heightens the cytotoxicity of alkylating agents and antimetabolites in cells [183,184]. A number of other proteins which play essential roles in DDR and which can be targeted for the development of more drugs include the Ku complex, serving as a binding platform for NHEJ proteins; Rad51 is a ssDNA binding protein and recombinase that plays a crucial role in strand invasion and the homology search process in HR; DNA pol θ, with the polymerase activity necessary for alt-EJ gap filling; TFIIH, a core protein in the NER pathway; and XRCC1, a central scaffold for enzyme complex assembly in BER and SSBR [185].

The significance of a specific DNA anomaly, known as covalent DNA–protein crosslinks (DPCs), has only garnered attention in the past decade [186]. DPC formation is a common phenomenon within cells and can be triggered by a range of factors, both endogenous, such as aldehydes generated during cellular metabolic processes, and exogenous, including ionizing radiation, ultraviolet light, and chemotherapeutic agents [187]. In contrast to other types of DNA damage, DPCs can be generated by any nuclear protein in close proximity to DNA, rendering them substantial and highly toxic. This toxicity is capable of disrupting nearly all chromatin-based processes [187]. DPCs are comprised of DNA, proteins, and the covalent bonds between them, resulting in variations among these three DPC components. Consequently, the repair of DPCs involves multiple repair pathways, presenting potential opportunities for novel combination therapy strategies in cancer treatment (see reviews [186,188]).

## 4. Conclusions

With the development and application of chemotherapy, drug resistance has become a challenging issue in cancer treatment. Therefore, in-depth research into the mechanisms of resistance to chemotherapy could improve the treatment prospects for patients. A profound understanding of aberrant DNA damage pathways’ roles in tumorigenesis and drug resistance is crucial. Furthermore, research into the mechanisms of genomic instability and DNA damage repair is an emerging focal point in basic cancer research and clinical treatment studies. Comprehending the hierarchy, redundancies, and interactions among distinct repair mechanisms will facilitate the development of synthetic lethality approaches aimed at specifically targeting malignant cancer cells. In many types of cancer, the ability to effectively respond to DNA damage is often lost. Utilizing antitumor drugs that block specific DNA repair pathways can enhance cytotoxicity, reverse drug resistance, and improve treatment efficacy. To enhance treatment effectiveness, DDR inhibitors can be used in combination with other drugs targeting DDR proteins or entirely different signaling pathways, with the aim of blocking the multiple pathways on which cancer cells rely for survival. After years of research, various inhibitors related to DNA damage response pathways have been developed. Additionally, DDR inhibitors can be combined with standard therapeutic drugs, such as in the use of PARP inhibitors to enhance the effectiveness of platinum-based drugs and in the evaluation of other studies involving the combined use of DDR inhibitors (including CHK1/2 and WEE1 inhibitors) with radiotherapy and chemotherapy. In the future, precision targeted therapy is expected to gradually replace chemotherapy and its stronger side effects, thereby improving the survival time and quality of life for late-stage cancer patients.

## Figures and Tables

**Figure 1 ijms-24-15907-f001:**
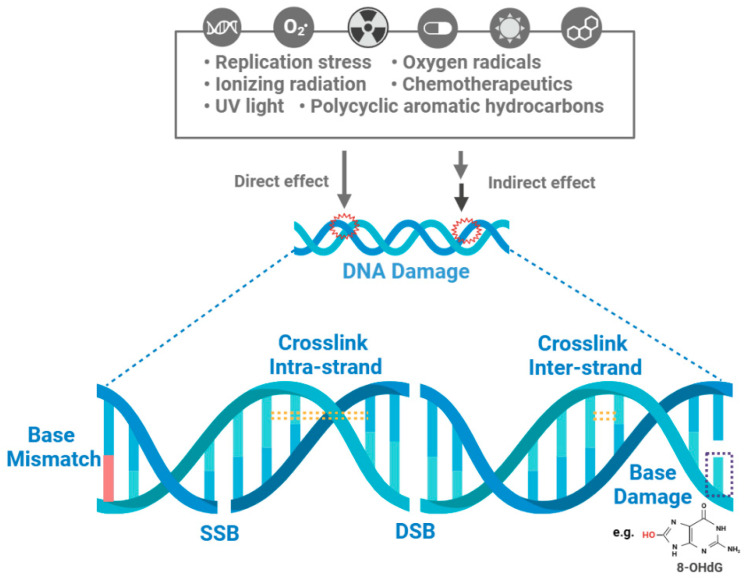
Types of DNA damage. DNA can undergo various types of damage due to both endogenous factors (such as replication stress and oxygen radicals) and exogenous agents (such as ionizing radiation, chemotherapeutics, UV light, and polycyclic aromatic hydrocarbons). These factors can interact with DNA directly or indirectly, resulting in the cleavage of chemical bonds within DNA molecules, ultimately causing changes in the structure and functionality of DNA. A wide array of DNA damage types exist, encompassing single-strand breaks (SSBs), double-strand breaks (DSBs), base damage (e.g., 8-oxoG), base mismatches, DNA crosslinks (both intra-strand and inter-strand), and more.

**Figure 2 ijms-24-15907-f002:**
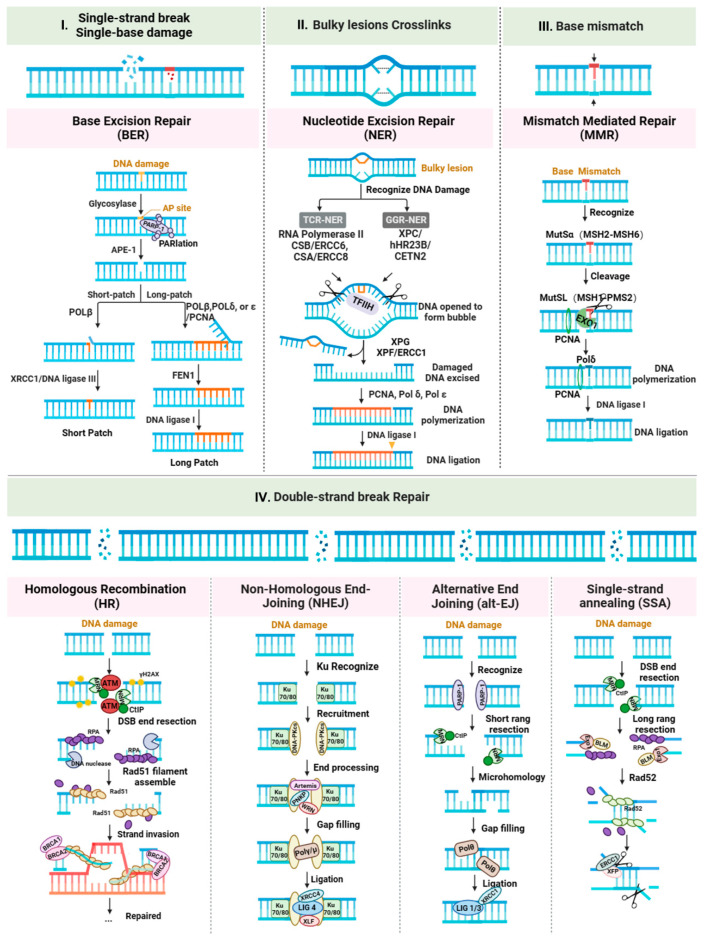
Overview of DNA damage repair pathways. **I**. Repair of single-strand break and single-base damage through direct and indirect base excision repair (BER). **II**. Repair of DNA adducts via transcription-coupled nucleotide excision repair (TC-NER) or global genomic nucleotide excision repair (GG-NER). **III**. Base mismatch repair via mismatch-mediated repair (MMR). **IV**. Double-strand break repair via homologous recombination (HR), non-homologous end joining (NHEJ), alternative end joining (alt-EJ), and single-strand annealing (SSA).

**Figure 3 ijms-24-15907-f003:**
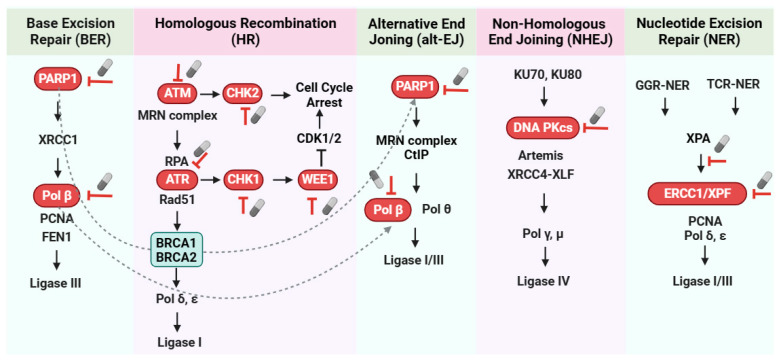
Targeting DNA damage response for cancer therapy. Inhibitors of DNA repair pathways, such as PARP1 (inhibits BER and alt-EJ pathways), Polβ (inhibits BER and alt-EJ pathways), ATM/ATR (inhibits HR pathway), and DNA-PKcs (inhibits NHEJ pathway), and the disruption of interactions between XPA and ERCC1 or ERCC1 and XPF (inhibits NER pathway). Inhibiting these proteins increases DNA damage and genomic instability in tumor cells. Checkpoint kinase inhibitors, such as Chk1/2 inhibitors and WEE1 inhibitors, abrogate cell cycle checkpoints, leading to mitotic catastrophe in tumor cells with high DNA damage. *BRCA* genes are integral components of the HR pathway and are relevant in synthetic lethality pathways. In the context of *BRCA* gene mutations, cells rely on alt-EJ pathways, including those involving PARP, to repair DNA damage. Therefore, in the presence of *BRCA* gene defects or mutations, PARP inhibitors lead to the accumulation of DSBs, subsequently causing cell death. Symbolic pills show potential targeted therapeutic interventions. BER: base excision repair; HR: homologous recombination; NHEJ: non-homologous end joining; alt-EJ: alternative end joining; NER: nucleotide excision repair.

**Table 1 ijms-24-15907-t001:** A series of inhibitors of DNA damage response (DDR) involved in Part 3.

Target	Inhibitor	Impair Pathway	Cancer/Cells	Current Status *
PARP1/2	olaparib	BER, alt-EJ	Advanced-stage and/or recurrent solid tumors with germline BRCA1/2 mutations	FDA-approved
	talazoparib	BER, alt-EJ	Advanced-stage and/or recurrent solid tumors with germline BRCA1/2 mutations	FDA-approved
	niraparib	BER, alt-EJ	Advanced-stage and/or recurrent solid tumors with germline BRCA1/2 mutations	FDA-approved
	rucaparib	BER, alt-EJ	Advanced-stage and/or recurrent solid tumors with germline BRCA1/2 mutations	FDA-approved
	velaparib	BER, alt-EJ	Advanced-stage and/or recurrent solid tumors with germline BRCA1/2 mutations	Phase III
ATM	KU-55933	HR	Myeloma cells, myeloid leukemic cells	
	KU59403	HR	Colon cancer xenografts	
ATR	M6620 (VX-970 or berzosertib)	HR	Solid tumors	phase I
	M4344 (VX-803)	HR	Solid tumors	phase I
	AZD6738	HR	Solid tumors	Phase II
	BAY1895344	HR	Solid tumors and lymphomas	phase I
DNA-PK	Nedisertib	NHEJ	Solid tumors	phase I/II
	AZD7648	NHEJ	Solid tumors	phase II
	VX-984	NHEJ	Solid tumors	phase I
CHK1/2	UCN-01	HR	Solid tumors	Phase II
	Rabusertib (LY2603618)	HR	Solid tumors	phase I
	MK-8776 (SCH 900776)	HR	Solid tumors and lymphomas	phase I
WEE1	Adavosertib (AZD1775)	HR	Solid tumors	phase I/II
DNA Pol β	pro-14	BER, alt-EJ	HeLa cells	
ERCC1	UCN-01	NER	A549 cells	
	NERI01	NER	Colon cancer cells	
	compound 10	NER	Colon cancer cells	

* More details about inhibitors can be obtained from ClinicalTrials.gov. BER: base excision repair; HR: homologous recombination; NHEJ: non-homologous end joining; alt-EJ: alternative end joining; NER: nucleotide excision repair.

## Data Availability

Not applicable.

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
