# Peer review of "Targeting the DNA Damage Response for Cancer Therapy"

_ijms, 2023, doi:10.3390/ijms242115907_

Round 1

Reviewer 1 Report

Comments and Suggestions for Authors

Wang et al. summarize the main DNA-repair mechanisms and the involvement of PARP1 in recognition and DNA damage response. The review is carefully written and contains all relevant information.

This reviewers have only a comments:

-       There are overlapping, alternative and/or redundant DNA repair pathways. However, the timing in the cell cycle sometimes the differences (replication, availability of DNA repair proteins, etc.). Maybe this could be highlighted in the context of synthetic lethality.  

-       DNA interstrand crosslinks are mainly repaired by the FA/BRCA-pathway (not discussed)

-       A table of the reviewed anti-cancer drugs with additional information regarding target tissues, affected pathway(s), state-of-use, etc. would be helpful.

Reviewer 2 Report

Comments and Suggestions for Authors

The manuscript reviews the various routes for therapeutic intervention targeting DNA damage response. The authors review the types of DNA damage and the repair pathways. Next, the authors review therapeutic targets and their roles in anti-cancer therapy. The authors did an excellent job of addressing many of the current DNA damage types. Furthermore, the graphics generated for the paper do an excellent job summarizing the sections covered. Overall, this is an excellent review of the subject matter, with many good references for the reader to look further into. One suggestion I might make is to separate the nucleotide crosslinking from protein DNA crosslinking (DPC). A couple of exciting strategies have come to light about how the cells deal with DPC and might be of interest for therapeutic strategies.

Otherwise, I feel that the review did an excellent job of addressing the current trends in DNA damage response and the current targets for therapeutic intervention.
